# OpenReview forum: "A3: an Analytical Low-Rank Approximation Framework for Attention"
_ICML.cc/2026/Conference — ICML 2026 regular_

### Official Review · Reviewer_tPXQ · 2026-03-05

**Soundness:** 3
**Presentation:** 2
**Significance:** 2
**Originality:** 3
**Overall Recommendation:** 4
**Confidence:** 5

**Summary:**

The paper proposes A3,  a post-training low-rank approximation framework. A3 splits a Transformer layer
into three functional components, namely QK, OV, and MLP, and provides analytical solutions that reduce the hidden dimension size inside each component while minimizing the component’s functional loss. The experimental results compared to SVD-LLM show the effectiveness of A3.

**Compliance With Llm Reviewing Policy:**

Affirmed.

**Final Justification:**

I raise my score.

**Key Questions For Authors:**

See weaknesses above.

**Limitations:**

See weaknesses above.

**Strengths And Weaknesses:**

Strengths:

- The paper is technically sound.

- The superiority to SVD-LLM shows the efffectivness of A3.

Weaknesses:

- First of all, this paper is much more like a channel pruning method rather than a LoRA-based method. The authors reduce the hidden dimensions of QK, OV and MLP using SVD decomposition and discard the unimportant channels (if I understand the motivation correctly, since the claim in the introduction section is a little bit misleading). Therefore, it is important for the authors to compare their methods with channel pruning methods, which are missed in the experiment section.

- The paper lacks competitors. You only compare with the baseline method SVD-LLM, and no SOTA LoRA/pruning methods are compared.

- What is the training time to compute your SVD method, compared to the original SVD method?

- Lack of actual accuracy test on downstream tasks, such as the commonly used commonsense reasoning datasets.

---

> ### Author Rebuttal · Authors · 2026-03-31
>
> We thank the reviewer for leaving comments and suggestions. We would like to address your concerns one by one.
>
> > W1
>
> Thank you for the question. We would like to first clarify that **$A^3$ is not a LoRA-based method** and we did not claim it is a LoRA-based method in the paper; rather, it is a training-free low-rank approximation approach. We decompose a transformer into its core components—QK, VO, and MLP—and compress each by minimizing its respective error (i.e., attention scores, attention patterns, and MLP outputs) under low-rank constraints. For QK and VO, we derive closed-form solutions based on **SVD decomposition**, while for the MLP we employ a heuristic approach based on CUR decomposition.
>
> Importantly, $A^3$ requires only a single autocorrelation matrix from the activations of each component within a layer to compute these solutions. No training or fine-tuning is needed.
>
> $A^3$ follows the line of work on training-free low-rank compression methods such as SVD-LLM[1], Palu[2], and Clover[3], rather than LoRA-based approaches. While our method reduces the hidden dimensions of QK, OV, and MLP, it does so by merging them into a unified weight matrix and performing low-rank approximation on this merged representation. This yields low-rank QK, OV, and MLP components, all grounded in principled low-rank approximation techniques. For QK and OV, the rank of each group component corresponds to the head dimension, whereas for the MLP it corresponds to the intermediate dimension.
>
> **We agree that structure pruning methods can be adapted to the $A^3$-MLP setting.** In Section 4.2, we compare against structure pruning methods such as Wanda[4] and abs(w). Additionally, Table 3 includes a comparison between $A^3$ and Clover[3], which is itself a structure pruning method.
>
> We will highlighting more explicitly how A³ relates to existing structured pruning approaches for MLP in the revised version.
>
> > W2
>
> Beyond SVD-LLM, we also compare $A^3$ with Palu (ICLR2025) and Clover (ICML2025) which are state-of-the-art low-rank compression and pruning methods. We kindly invite you to revisit Section 4.2 and Table 3, where these comparisons are presented.
>
> > W3
>
> | Model|SVD (s/layer)|$A^3$ (s/layer)| SVD / $A^3$|
> |-|-:|-:|--:|
> | LLaMA-3.1-8B| 5.21 | 34.80 | 0.15x|
> | LLaMA-3.3-70B| 29.71| 254.29| 0.12x|
>
> Table A1: Compression time per decoder layer (seconds)
>
> $A^3$ is a training-free method. Here, we compare the time required to compute low-rank approximations against a simple SVD applied directly to model weights.
>
> $A^3$ is significantly slower on a per-layer basis due to the eigendecomposition involved in computing $\sqrt{R_{xx}}$, particularly for the FFN component. For example, $R_{xx}$ has dimensions 14,336 × 14,336 in 8B models and 28,672 × 28,672 in 70B models. This operation dominates the overall runtime of $A^3$. In contrast, the SVD baseline avoids this cost entirely, as it operates directly on the weight matrices.
>
> In terms of full-model runtime, for an 8B model (32 layers), SVD takes approximately 167 seconds, while $A^3$ requires around 1,114 seconds. For a 70B model (80 layers), SVD takes about 2,377 seconds (40 minutes), whereas $A^3$ takes approximately 20,343 seconds (5.6 hours).
>
> It is important to note, however, that although simple SVD is faster, the resulting compressed models are already broken. In contrast, $A^3$, while slower, performs compression as a one-time offline process and produces substantially better results.
> > W4
>
>
> In Table 2, we evaluate our method on several widely used commonsense reasoning benchmarks, including ARC-C, BoolQ, WinoGrande, GSM8K, and MMLU.
>
> These datasets are standard for assessing reasoning capabilities in large language models. In particular, MMLU covers 57 subjects across STEM and the social sciences, providing a broad evaluation of both world knowledge and problem-solving ability. GSM8K consists of 1.32K math word problems designed to test multi-step mathematical reasoning.
> These results demonstrate that $A^3$ generalizes well to diverse downstream reasoning tasks.
>
> Thank you once again for leaving comments and suggestions. We hope this address your concerns and we look forward to further discussing any additional points you may have.
>
> ## Reference
> [1] SVD-LLM: Truncation-aware singular value decomposition for large language model compression. ICLR 2025
>
> [2] Palu: Compressing kv-cache with low-rank projection. ICLR 2025
>
> [3] CLOVER: Cross-Layer Orthogonal Vectors Pruning. ICML 2025
>
> [4] A simple and effective pruning approach for large language models. ICLR 2024.

---

> > ### Author Rebuttal · Reviewer_tPXQ · 2026-04-01
> >
> > Thanks for the author's rebuttal. My concerns are solved, and I will raise the score from 3 to 4.

---

### Official Review · Reviewer_7UXF · 2026-03-09

**Soundness:** 3
**Presentation:** 3
**Significance:** 2
**Originality:** 2
**Overall Recommendation:** 4
**Confidence:** 3

**Summary:**

This paper integrates in a recent workstream on compressing LLMs with low-rank approximations. While the vast majority of previous works in this domain focuses on decomposing individual linear layers, the authors take a different approach and target more native components of the transformer architecture, namely the pre-softmax attention score (“QK”), the attention output (“VO”), and the feedforward activations (“MLP”). In this way, the low-rank “bottleneck” dimension directly corresponds to an (interpretable) hidden dimension of the transformer. Therefore, compared to layer-wise factorization, no additional matrix multiplication operation is introduced, which is desirable for efficient inference. Algorithmically, the decompositions for the attention components are inspired by recent work on SVD approximations in LLMs (SVD-LLM), whereas the MLP component is based on a CUR approximation. The authors support the effectiveness of their method by evaluating common base models on standard benchmarks in the model compression literature (perplexity & LM Eval downstream tasks). In their ablation studies, they also demonstrate that their algorithm can be combined with quantization to achieve even lower compression rates.

**Compliance With Llm Reviewing Policy:**

Affirmed.

**Final Justification:**

I acknowledge that the authors did a genuine effort in responding to my review and I will increase my score to a weak accept as promised. My concerns about practical significance still remain, but the paper is overall sound and the method might be interesting to the research community.

**Key Questions For Authors:**

1. Is A^3 compatible with FlashAttention?
2. Isn’t the A^3-MLP component basically a structured pruning approach, rather than a low-rank decomposition? If yes, this link should be made in the paper.
3. Regarding L211 left: In standard transformers, the head dimension d\_qk is typically relatively small (depending on \#heads), so the matrix W\_qk is already quite low-rank. Maybe I missed an important point here, but doesn’t this have a negative impact on the compressibility compared to “denser” individual layers?
4. Regarding the fine-tuning experiments in Appendix D: Did you perform OOD tests (e.g., on LM Eval tasks) to evaluate overfitting? Are the LoRA adapters mergeable with your low-rank matrices? Otherwise this could have a negative impact on the inference throughput.

**Limitations:**

yes

**Strengths And Weaknesses:**

While I acknowledge the originality of the proposed A^3 approach, and that it addresses some relevant problems in the field of low-rank approximation, I have several concerns about the significance of the results and scope of the experimental study. Please find my detailed comments below.

### Strengths

1. Overall, the paper is technically sound. While the scope of the experimental study is limited (see below), the authors provide evidence for their main claims.
2. The presentation of the paper is good. The text is well-structured and easy to follow. My concerns about the presentation are minor and addressable, see “Minor” below.
3. While building on techniques and insights of previous works in the field, to my knowledge, the key idea of applying low-rank approximation to the native architectural components of transformer models is original and interesting. A^3 addresses a few relevant issues of the predominant layer-wise approximation paradigm:
   1. By reusing the existing components of the transformer architecture, no inference overhead is introduced by an extra GEMM with second low-rank factor. And through the proposed compression method, additional speed gains can be expected, as the authors demonstrate empirically.
   2. The standard layer-wise low-rank compression suffers from a “doubling issue”: When initially representing a (square) weight matrix W via SVD, the number of parameters is effectively doubled\! This is not only problematic for inference speed (see the previous point) but also for compressibility. Indeed, to achieve any compression, half of the singular values need to be pruned, which may already lead to a loss of accuracy. A^3 does not suffer from this issue. Interestingly, the authors do not explicitly point out this notable advantage of A^3 in their work. I would encourage them to do so because it is in favor of their method.
   3. With the attention-related components of A^3, it is possible to implement a KV-cache compression technique, which is interesting in its own right.
4. The compatibility of A^3 with modern GQA and RoPE is appreciated. However, it is worth pointing out that the need for such an extension is specific to the considered method. Not all previous works on layer-wise approximation do severely interfere with GQA and RoPE and may be applied with less “care” – so I disagree with this claim in L78 left in full generality: *“This overcomes the limitation of existing low-rank approximation methods, which can only be applied on the vanilla MHA architecture.”* If you have specific limitations in mind, please point them out.
5. I acknowledge the series of ablation studies presented in the main body and Appendix D. In my view, the most interesting one is the combination with quantization, because it points to a general practical use case of low-rank approximation methods for LLMs, see my weak points below for an additional comment on that.

### Weaknesses

1. My most important concern is about the significance of the method and experimental study:
   1. My overall conclusion from the numerical experiments is that A^3, as a standalone method, only works decently for very low compression rates (\<20%). Here, my expectation is NOT that a low-rank approach can compete with SOTA quantization techniques (this is a completely different domain and the authors have demonstrated the complementary nature). Nevertheless, the analysis is strongly focussed on perplexity, which is known to be only a rough proxy of cognitive capabilities of LLMs. In my view, the results of Table 2 on downstream tasks do not look promising. For example, the LLaMA-3.1-70B at only 20% compression already performs worse than the LLaMA-3.1-8B base model. These results do not make hope that A^3, as a standalone approach, can lead to very competitive compression results.
   2. While my previous point is more or less a statement about low-rank compression algorithms for LLMs in general (which for sure have their raison d'être), my expectation for the present paper is that the authors show that A^3 can compete (accuracy-wise) with existing low-rank approximation SOTA that target individual layers. While SVD-LLM apparently is still a popular baseline, it is not SOTA anymore. It therefore remains unclear how A^3 would compete with already published follow-up work like Dobi-SVD or ACIP; see references below.
   3. Moderate: I was quite surprised to see in Figure 7 that compressing larger models (70B) is so expensive and if I understood correctly this is specifically due to how A^3 compresses the attention components of the model.
   4. Moderate: I’m not sure if the dependence of A^3 on calibration data is desirable, as reported in Appendix D. Calibration data is a remedy to achieve better “recovery” after compression, but it always comes with the danger of overfitting and the author’s findings provide evidence in that respect. This becomes even more problematic with fine-tuning (see one of my questions below).
2. The inference study does only report results on prefill and I think it should cover decoding as well because the potential speed gains of A^3 are a key feature. On the other hand, in the relevant compression regime of \<20% (where the model is still functional), the gains are only relatively small (see Figure 3).
3. Moderate: The results on combining A^3 with quantization in Appendix D are a welcome addition and indicate where low-rank methods have the most practical potential, namely as a complementary “booster” of other compression techniques. However, I think this study has several weak points and could be extended:
   1. The orthogonality of quantization and low-rank approximation is not a new finding and has been demonstrated in existing works. Since A^3 does not target individual layers, it would be good to compare low-rank+quantization with other baselines like SVD-LLM so see if A^3 works comparably well.
   2. Figure 5b: How are the x-values for HQQ being computed?
   3. What group sizes are used for HQQ? This can have a significant impact on the compression quality and size reduction. In particular, the overhead due to groups is not taken into account in Table 10 and might lead to (slightly) different conclusions.
4. Moderate: While A^3 addresses some relevant issues of existing low-rank approaches (see “Strengths”), the price is that it is strongly tailored to the transformer architecture and therefore requires special care for advanced components like GQA and RoPE. This limits the generality and flexibility of the approach.
5. Moderate: It would be good to see results on more recent models and other model families like Qwen3 or Mistral.

#### References

*Qinsi, W., Ke, J., Tomizuka, M., Keutzer, K. and Xu, C., Dobi-SVD: Differentiable SVD for LLM compression and some new perspectives. In The Thirteenth International Conference on Learning Representations, 2025\.*

*Genzel, M., Putzky, P., Zhao, P., Schulze, S., Mollenhauer, M., Seidel, R., Dietzel, S. and Wollmann, T., Choose Your Model Size: Any Compression of Large Language Models Without Re-Computation. Transactions on Machine Learning Research, 2025\.*

### Minor Issues

1. Typos:
   - Equation (9): Sub-indices “q” and “kv” are missing at X
   - Table 4: Sub-indicies in Line 589ff
2. Some of the figure captions could be more detailed. For example: What does 0k correspond to in Figure 3? What is the compression ratio in Figure 4? Or what is the x-axis for HQQ in Figure 5b? In general, please try to make the figure captions as self-contained as possible.
3. Table 6 is not well located. I initially thought that it is related to quantization, but this is obviously not the case.
4. I find the A^3-QK part of Figure 1 a bit misleading. One would think that W\_q and W\_k are simply truncated but they are actually replaced by low-rank factors, right?
5. The original base model (BF16) performance should be reported in Table 1 and Figure 2\.
6. I was confused when reading this in L285 right: *“This section presents the main evaluation results where we compare A3 against all the baselines that can be applied to all of the three main components (QK, OV, MLP) in Transformer.”* I think the authors report the numbers from the SVD-LLM paper, but this phrase sounds like it is somehow adapted to the (QK, OV, MLP) component of the A^3 approach. Please try to avoid such ambiguities by clearly writing how the baselines are applied and from where the scores origin (in-house or literature).
7. While the number of parameters is a widely used measure for the size of a (compressed) model, it would be also important to report allocated VRAM, at least in some of the experiments. This is particularly important in the context of the quantization experiments in Appendix D because quantization does not reduce the number of parameters but their precision.
8. Caption of Table 3: *“\[...\] CRatio indicates compression ratio on both KVCache and parameter count.”* What does this precisely mean?
9. Reporting four digits in Table 2 is meaningless given typical standard deviations in LM Eval runs.
10. Calling the theoretical results in Section 3.1 “Theorems” is a bit over-stated.

---

> ### Author Rebuttal · Authors · 2026-03-31
>
> Thank you for the valuable suggestions. We would like to answer your questions one by one.
>
> >W1.1
>
> Please refer to W.2 in Reviewer RoK3
> >W1.2
>
> Before presenting our results, we would like to clarify that the comparison between Dobi-SVD, ACIP, and A³ is not strictly one-to-one. A³ is fully training-free, whereas Dobi-SVD and ACIP rely on training to optimize rank allocation and mitigate truncation error, giving them an inherent post-hoc error compensation, and optimizing rank allocation is out of the scope of this paper. In this paper, we simply use uniform rank throughout our evaluation.
> In contrast, A³ is memory-efficient, requiring only a single $R_{xx}$ matrix per component in each decoder, which reduces storage and deployment complexity. For a fair comparison, we follow ACIP and report results both with and without LoRA-based error compensation, as introduced in ACIP.
>
> [Table 19: Comparison with ACIP, SVD-LLM, A3.](https://bashify.io/i/BPUf6h)
>
> Without LoRA correction, A³ consistently outperforms both ACIP and SVD-LLM across compression ratios. However, as compression becomes more aggressive, ACIP with LoRA begins to provide stronger error compensation and eventually surpasses the training-free A³ variant.
>
> Additionally, we want to highlight that A³ could benefit from integrating penalty based rank truncation from ACIP, penalising the diagonal term between QK, VO through training, leading to optimal global rank selection. We leave this hybrid direction as promising future work. In the revised manuscript, we will further clarify the relationships among A³, ACIP, and Dobi-SVD, and update the related work section to explicitly include both ACIP and Dobi-SVD.
> >W1.3
>
> Please refer to W.3 in Reviewer tPXQ.
> >W1.4
>
> Please refer to Q.1 in Reviewer RoK3.
> >W.2
>
> Here we provide result on decoding runtime speedups across batch sizes, prefill & decode token lengths.
>
> [Table 18. Decoding throughput for LLaMA-2-13B.](https://bashify.io/i/zi3NWq)
>
> Since A³ does not alter model architecture and only reduces linear layer dimensionality, it remains fully compatible with existing inference stacks (Appendix E, we evaluate both Eager and SDPA (FlashAttention) kernels), achieving consistent decoding speedups.
> In contrast, methods like SVD-LLM introduce extra kernel launch overhead, increasing latency. We will add decoding-time analysis in Appendix E.
> >W3.1
>
> We agree that the orthogonality between quantization and low-rank has been studied in prior work, and we do not claim this as a novel finding in our paper. Instead, our goal is to empirically verify if this property also holds in the context of A³. Here, we report A³/SVD-LLM+quantization on WikiText-2 PPL using LLaMA-3.1-8B with 4-bit HQQ (gs 64) quantization, matching compression by ensuring both methods reduce an equivalent number of parameters.
>
> [Table 16. PPL on WikiText-2](https://bashify.io/i/m3vSEp)
>
> SVD-LLM + quantization significantly degrades model quality, while A³ remains substantially more robust and maintains lower perplexity across compression ratios under the same setting.
>
> >W3.2, W3.3
>
> In HQQ, 8-bit zero point and 16-bit scale are amortized over each group:
> $$
> \text{bits per weight} = \frac{24}{\text{group size}}
> $$
> With group size 64, this adds 0.375 bits per weight. All 4-bit experiments use group size 64. We will clarify this configuration in the appendix.
>
> > W.4
>
> Modern transformers utilize NoPE, RoPE, MHA, GQA and three layer MLP. $A^3$ supports all of them, while Palu and Clover don't support RoPE, introducing inference cost.
> >W.5
>
> Here is Mistral-3-3B and 8B results in Table 1. We can see A3 still works really well. We will this to the revised version.
>
> [Table 17. Ministral-3s in Table 1](https://bashify.io/i/x2ufIY)
> >Q.1
>
> Please refer to W.2 reply.
>
> >Q.2
>
> Yes, structured pruning method can be adapted to A³-MLP. As shown in Figure 4, A³ is compared with structured pruning methods like Wanda and magnitude pruning.
> We will highlighting how A³ relates to existing structured pruning approaches for MLP.
> >Q.3
>
> In A³, we reduce $d_{\text{head}}$ for QK, VO, which we find remains highly compressible. On MPT-7B, about 40% of rank in QK/OV can be removed with minimal degradation (Table 11), indicating substantial redundancy even in small head dimension.
>
> Unlike standard low-rank methods constrained by $\frac{(m+n)r}{mn}$ scaling, A³ leverages a pre-structured low-rank form of $W_{qk}$, so reducing rank by r% directly translates to proportional savings in parameters and FLOPs without SVD reparameterization overhead.
>
> This allows A³ to better exploit redundancy than layerwise low-rank approaches.
> >Q.4
>
> We follow SVD-LLM fine-tuning setting with Alpaca-cleaned samples. Alpaca is instruction-following QA and out-of-distribution relative to WikiText-2 eval.
>
> Since A³ only reduces linear layer dimensionality without altering architecture, LoRA is fully mergeable.
>
> We will use the second rebuttal to address your minor issues due to the strict word limit. Thank you!

---

> > ### Author Rebuttal · Reviewer_7UXF · 2026-04-03
> >
> > I thank the authors for their detailed response which addresses most of the raised issues. With this, I am willing to adjust my overall recommendation from 3 (weak reject) to 4 (weak accept) provided that the authors address the minor issues in detail as promised in their response. I will not go beyond this rating, however, as my main concerns about significance remain: LLaMA-3.1-70B at only 20% compression already performs worse than the LLaMA-3.1-8B base model. It therefore remains questionable whether $A^3$ can lead to any practically useful compression gains that justify using a larger model compressed with $A^3$ over using a smaller model.

---

> > > ### Author Response · Authors · 2026-04-03
> > >
> > > Thank you for the detailed review and the positive acknowledgment of our previous rebuttal, as promised, here we address the rest of the minor limitations one by one.
> > > > M.1
> > >
> > > Thank you for pointing this out. We will revise the manuscript to add subscripts in Eq.(9) and fix typos in Table 4 588-590 notations.
> > > > M.2
> > >
> > > In Figure 3, 0k means 0 delta TPS as no improvement compared with the full model Eager or SDPA implementation.
> > > In Figure 4, we compare all components with compression ratio = 20%. We state this in its related paragraph and will add this to the caption.
> > > In Figure 5, x-axis is the overall compression ratio, we will add an x_label to the graph.
> > > > M.3
> > >
> > > Yes, we will relocate Table 6 to rank allocation paragraphs.
> > > > M.4
> > >
> > > We agree with you. We will introduce visualisations that present the notion of low-rank factorization step before we present the low-rank truncation workflow.
> > > > M.5
> > >
> > > We didn’t include them due to the page limit. We will add a row in Table 1 and a horizontal line in Figure 2 with a caption to indicate the original base model performance.
> > >
> > > > M.6
> > >
> > > Here we mean all linear layers are compressed. Yes, we borrowed the results from the SVD-LLM paper and adapted A3 using identical settings in SVD-LLM (e.g calibration data set and size). We will clarify this clearly in the preliminary experiments paragraph.
> > > >M.7
> > >
> > > In Table 12, we report both the theoretical and actual peak VRAM usage for A3 at compression ratios of 20–60%, as shown in the “Peak Memory” column. We will revise the table columns to more explicitly represent VRAM usage.
> > >
> > > > M.8
> > >
> > > Since Table 3 compresses the QK and VO components by reducing their head dimension, additionally the size of parameters, the size of the KV cache (head_dimension × num_layers × sequence_length × num_heads), being directly proportional to the head dimension, decreases accordingly. Therefore, a 20% reduction in head dimension results in a corresponding 20% reduction in both the weights and the KV cache size. Details shape analysis can be found in Appendix E.1. We will clarify this statement in the revised manuscript.
> > >
> > > > M.9
> > >
> > > Thank you for the suggestion, we’ll limit it to two digits to make the result more focused and easier to read.
> > >
> > > > M.10
> > >
> > > Thanks for your suggestion. We will reframe Section 3.1 to avoid over-statement.
> > > We used ‘theorems’ to structure the method section in the form of problem definition - lemma reformulation - theorems, which differentiate from heuristic solutions.
> > >
> > > Thank you once again for leaving these suggestions. We hope this address your minor issues and we look forward to further discussing any additional points you may have.
> > >
> > > > Updates on 7th April
> > >
> > > We thank again the reviewer for the time invested in raising the questions and comments that improved the quality and scope of our work. Following the reviewer's suggestion to raise the score to 4, we kindly ask if she/he can validate this change within the platform before the deadline.

---

### Official Review · Reviewer_RoK3 · 2026-03-11

**Soundness:** 4
**Presentation:** 4
**Significance:** 3
**Originality:** 3
**Overall Recommendation:** 4
**Confidence:** 4

**Summary:**

The paper introduces A3 -- a method to build low-rank approximations of three Transformer components independently: QK, OV and MLP. The solution is build with applications to Transformer in mind, thus it is optimizing the layer output error instead of a typical matrix error as vanilla SVD baseline methods do. The paper provides solid mathematical justifications and extensive experiments. As a result, the method results in a significant improvement over current SoTA.

**Compliance With Llm Reviewing Policy:**

Affirmed.

**Final Justification:**

The paper introduces a method to build low-rank approximations of three Transformer components independently: QK, OV and MLP. The paper is easy to read and follow, and the paper provides extensive experiments. It does have some limitations: limited compression rate, which is however improved in the setups without RoPE module or if combined with other methods. On another hand, most of the modern architectures do utilise RoPE. I am keeping the Weak Accept score since I believe it adequately describes this submission: interesting work, but with some limitations in applicability.

**Key Questions For Authors:**

Q1: Matrix $R_{XX}$ -- autocorrelation matrix of calibration activations -- is a key component of the solution. You mention in Appendix C the size of the set -- 128 sequences of 2048 tokens. Have you tried other values? What happens if you increase or decrease the calibration size?
Q1.2: What would happen if calibration dataset has different distribution from the benchmarks? From Table 7 it seems that with lower compression setup, the method is robust to the choice of the dataset, but with a higher compression, the gap becomes larger. Is it due to common instability of low rank methods? What is your hypothesis on this?

Q2: MLP components contains nonlinearity and thus becomes trickier to tackle. What happens if we do not compress it? Obviously, the overall compression will be less impressive, but, for example, can it be compensated by more aggressive compression of QK/OV components?


Q3: I struggle a bit to understand the sentence on line 246 (_"This CUR approximation is a well-studied NP-hard problem.."_)
What does "This CUR approximation" refer to here? CUR approximation (aka matrix skeleton decomposition) is usually constructed by taking a subset of columns (C), rows (R) and its intersections forming matrix U; the columns and rows are selected from the same matrix which we want to decompose. In Equation 18 (the closest equation to the sentence), I do not see any matrix candidate for CUR decomposition. So my question here is: is there any specific matrix you are referring to on line 246 or is it just an introduction sentence for CUR decomposition?

Similarly, I am a bit confused by lines 283-284 (column 1): you are building a CUR decomposition for matrix $(R_{xx}^{1/2}W_{q,i})(W_{ki}^T R_{xx}^{1/2})$ and set $L=R_{xx}^{1/2}W_{q,i}$  and $R=W_{ki}^T R_{xx}^{1/2}$, but this is not a classical CUR decomposition, where you sample columns and rows?

**Limitations:**

yes

**Strengths And Weaknesses:**

Strengths:

- The paper presentation is excellent. The text is clearly written, it is easy to follow, mathematical derivations are clearly structured. Special thanks for Appendix A, which I found very helpful since notation becomes a bit too heavy by the end of Section 3.
- The paper provides rigorous derivations of the method, properly citing important and relevant works and building blocks of the method. This gives a reader a good understanding of the area landscape.
- The experimental section covers experiments on two families of LLaMA models (v2 and v3) with two model sizes within each and the Phi model. The model sizes span a wide range from 7B to 70B, which gives a great perspective on how the solution scales with growing model size. The evaluation was done by measuring both perplexity and downstream accuracy. The authors have compared the proposed method to the natural baseline -- low-rank SVD approximation. The authors also provide extensive runtime analysis (both theoretical and actual experiments in Table 12). Additionally, authors have discussed integration with quantization and LoRA training. Overall, experimental setup provides enough evidence to understand the method's performance.
- The method itself approaches low-rank approximation from a perspective of weight matrix being an actual part of the network, thus layer error output is a better target than a direct matrix approximation error. Since the method is built directly for Transformer architecture, the approach is different from vanilla SVD decomposition applied on a weight matrix. Motivated by these limitations, the proposed method takes this into account and advances the SVD based baseline. The method independently tackles compression of QK,OV and MLP components, providing an individual objective for each case.
- The problem tackled in the paper is important for the community since it addresses the growing size of LLMs limiting its usage and deployment in certain scenarios.
- The method results in improved accuracy compared to the baseline.

Weaknesses:
- Despite providing a set of ablations, I think the study could extend it a little bit further. For example, what happens if $R_{xx}$ matrix is replaced with identity (also check Q1).
- In general, the accuracy drop is quite significant, while a compression rate is 10-20%, which is somewhat modest. Thus, I am not sure how practical the method could be in isolation for individual practitioners. However, I acknowledge the stable improvement over the baseline and resulting improvement in throughput. Also, the paper provides evidence that A3 could be used as one of the building blocks composed together with other methods.

---

> ### Author Rebuttal · Authors · 2026-03-31
>
> Thank you for careful reviewing and raising questions about experiment. Here we answer them one by one.
> >W1
>
> Setting all $R_{xx}$ matrices to identity reduces our method to Clover, which shows sharp degradation even at mild compression (e.g., 20%, Table 3). Additionally, in Figure 4, we define A3-Q-only and A3-K-only by replacing $R_{x_qx_q}$ or $R_{x_kx_k}$ with identity and compare them to A3-QK. We provide a more detailed discussion of the relationship between Clover and these baselines in terms of the autocorrelation matrix in A3 in Section 4.2, and will further clarify this connection in the revision.
> >W2
>
> We attribute this limitation primarily to CUR decomposition, which can become unstable under aggressive compression (discussed in Limitation Appendix D). Table 3 shows that removing CUR in MLP layers yields significantly stronger compression–accuracy trade-offs.
>
> When CUR decomposition is necessary, we also explore fine-tuning with (A^3). As shown in Tables 9–10, even simple fine-tuning consistently improves performance, effectively mitigating degradation.
>
> Finally, as discussed in Appendix D, we believe a promising direction for practical deployment is a hybrid composition strategy combining complementary methods depending on module structure (e.g., A^3 for QK/VO components and SVD-LLM for MLP layers).
> >Q1
>
> Here we report results on varying the calibration set size beyond the default setting to study its effect.
> As shown Fig.A1, increasing the number of calibration samples consistently improves A^3 performance until it gradually saturates, indicating a stable convergence behavior.
>
> [Fig.A1: PPL vs num calibration samples](https://bashify.io/i/zrk830)
>
> Regarding Q1.2, Assume the calibration dataset is OOD. In this case, the SVD of $R_{xx}$ may not reflect true underlying data variance structure. If we had access to the true distribution’s autocorrelation matrix, and we projected it onto the basis derived from the OOD-estimated $R_{xx}$, the resulting coefficients would likely not exhibit a monotonic decreasing order with respect to rank, making rank truncation suboptimal.
>
> Now consider the case where both calibration and evaluation are performed on WikiText. In this setting, we can reasonably treat $R_{xx}$ as representing the true distribution. Any performance degradation from truncating to rank r can then be attributed purely to low-rank approximation error.
>
> In contrast, suppose we calibrate on SlimPajama and evaluate on WikiText. Here, SlimPajama serves as an OOD calibration dataset. We can analyze how well the basis derived from the OOD-estimated $R_{xx}$ explains the true WikiText autocorrelation structure. The projection coefficients are unlikely to follow a strictly decreasing order across ranks, implying that truncating to rank r leads to a suboptimal representation and consequently worse performance compared to the in-distribution calibration scenario. Leading to widen gaps across more aggressive compression.
> For practical use, we suggest a mixture of datasets for calibration. As shown in Table 8, using a more diverse calibration dataset can get us closer to the true training distribution, improving the performance of both A3 and SVD-LLM.
>
> >Q2
>
> Yes, the MLP components are challenging. We also noticed this and evaluated A3 without compressing MLP in Table 3. We observe even stronger gains at higher compression ratios. This is attributed to the closed-form solution that directly minimizes attention score and output errors, making it especially effective for attention components. We will clarify this design choice in the revision.
>
> >Q3
>
> In Eq.18, we treat $R_{xx}^{1/2}$ as C and $W_d$ as R matrix, the goal is to find the best r cols in $R_{xx}^{1/2}$ and the corresponding row in $W_d$ to approximate $R_{xx}^{1/2} W_d$. The matrix U to solve selects rows and columns from the original $R_{xx}^{1/2}$ and $W_d$.
> Line 246 is more of an introduction to CUR decomposition and the subsequence lines then illustrate how we formulate and apply existing CUR decomposition solutions to construct A3-MLP.
> Appendix B.3  shows how we derive the equivalent objective from problem 3 to lemma 3.4.
>
> In lines 283-284. We treat $R_{xx}^{1/2} W_q$ as C and $W_k^TR_{xx}^{1/2}$ as R. Note that RoPE requires adjacent dimensions for dot product attention, thus we cannot select arbitrary rows/columns (we have to select pairs of adjacent rows/columns). We followed [paper](https://epubs.siam.org/doi/abs/10.1137/S0097539704442684) (or [here](https://www.stat.berkeley.edu/~mmahoney/f13-stat260-cs294/Lectures/lecture02.pdf) for a shorter lecture notes version), which solves CUR by sampling rows/columns that corresponds to top-k
> $$\lambda_i := || C_{:, i} ||_2^2 \cdot || R_{i, :} ||_2^2$$
> (k is the truncation rank). To adopt it for RoPE, we sort by the sum of $\lambda_i$ of adjacent pairs of rows/columns and only keep the top-k/2 row/column pairs (See Appendix B.5 for the calculation of $\lambda_{2i}$).

---

> > ### Author Rebuttal · Reviewer_RoK3 · 2026-04-03
> >
> > I thank the authors for the provided rebuttal.
> > I keep the score since I believe that the paper is solid with interesting contribution but with some limitations mostly coming from a modest compression ratio and associated ppl drop, which might limit the applicability.

---

> > > ### Author Response · Authors · 2026-04-05
> > >
> > > Thank you for your response and for finding the work solid and interesting. Regarding the concern about modest compression, we would like to highlight two points: (1) our method can be effectively combined with quantization to achieve multiplicative compression gains, as detailed in the appendix D; and (2) for attention without RoPE, our approach achieves an 80% compression ratio while delivering perplexity an order of magnitude lower than Clover[1] and Palu[2], owing to its closed-form solution.
> > >
> > > Additionally, there is a growing body of work exploring the removal of RoPE from model architectures [3,4,5], and we believe A3 offers a valuable contribution within this emerging design. We hope this helps address your concern. We are open to address any further points that could improve your score of the work.
> > >
> > > [1] CLOVER: Cross-Layer Orthogonal Vectors Pruning and Fine-Tuning. ICML 2025
> > >
> > > [2] Palu: Compressing KV-Cache with Low-Rank Projection. ICLR 2025
> > >
> > > [3] Extending the Context of Pretrained LLMs by Dropping Their Positional Embedding. ICLR2026
> > >
> > > [4] Rope to Nope and Back Again: A New Hybrid Attention Strategy. NeurIPS 2025
> > >
> > > [5] Round and Round We Go! What makes Rotary Positional Encodings useful? ICLR 2025

---

### Official Review · Reviewer_hk78 · 2026-03-12

**Soundness:** 3
**Presentation:** 4
**Significance:** 3
**Originality:** 2
**Overall Recommendation:** 4
**Confidence:** 4

**Summary:**

A^3 is a post-training low-rank approximation framework for compression of Transformer-variant large language models (LLMs).  Unlike existing low-rank approximation techniques that produce A/B component matrices of rank r that approximate the full rank parameter matrix of a linear layer, A^3 leverages either the truncated singular value decomposition (SVD) or the CUR decomposition of the full rank weight matrix to reduce the hidden dimension of the model.  A^3 applies this approach not simply to the linear layers of the Transformer but to the individual components therein, including (i) QK component, (ii) the attention output, and (iii) the MLP output.  This approach achieves state-of-the-art performance across various benchmarks as well as increased computational and memory compression relative to other low-rank compression techniques.

**Compliance With Llm Reviewing Policy:**

Affirmed.

**Final Justification:**

No additional comments beyond what have been mentioned in other parts of the review.

**Key Questions For Authors:**

1.	Regarding Equation 9, the equation appears to omit the subscripts for the X terms, making the formulation somewhat ambiguous.

2.	In Section 3.3, beneath Equation 17, you state that U determines which columns of W_d to keep.  I believe you mean which rows of W_d / which columns of X_d to keep, correct?

3.	In Table 2, you present results demonstrating that under certain configurations, SVD-LLM outperforms A^3 on almost every tested benchmark.  You provide a plausible explanation for this in Appendix D, suggesting that SVD-LLM is overfitting.  Please provide more detail to support this hypothesis.  Your results in Table 8 do provide some evidence that overfitting may be at issue, but given that SlimPajama is utilized for calibration of the models tested in Table 2, why would SVD-LLM's overfitting result in improved performance across multiple benchmarks?  Do you have some arguments regarding correlation between those benchmarks and the calibration set that might explain these results?

**Limitations:**

yes

**Strengths And Weaknesses:**

Strengths:
1.	As compared to existing low-rank compression techniques, A^3 achieves state-of-the-art performance on various industry-standard benchmarks, demonstrating performance across various text distributions (e.g., WikiText-2, C4, and SlimPajama) and various down-stream tasks (e.g., multi-task knowledge retrieval, comprehension, and reasoning [mathematical and logical]).

2.	As compared to existing low-rank compression techniques, A^3 achieves superior compression (i.e., proportional to r, rather than ((m + n)/mn)*r) of both (a) computational complexity (i.e., by directly reducing the hidden dimension of approximated model parameters, rather than generating A/B low-rank component approximations, avoiding the need for additional matrix multiplications), and (b) memory overhead, due to the reduced computations resulting from smaller hidden dimensions.

3.	The authors present application of A^3 to several prominent variations of the standard Transformer framework, including other non-low-rank compression techniques, demonstrating its compatibility with these variations and its effectiveness when combined therewith.

Weaknesses:
1.	The approach proposed in this paper is an application of existing compression techniques to smaller sub-components of Transformers, which does not reflect significant novelty.

2.	Despite empirical results, the paper does not provide sufficient theoretical basis to explain why minimizing the error of the functional components of the Transformer should yield superior results to minimizing the error of layer output.

3.	While the paper provides some comparative benchmark performance results between A^3 and (i) the relevant base model, (ii) existing low-rank compression techniques, and (iii) existing pruning techniques, it does not provide comparative results showing (a) relative benchmark performance of A^3 and existing quantization techniques or (b) relative computational/memory compression of A^3 compared to existing pruning and quantization methods.

4.	Although the general performance of A3 is shown to be superior to that of other low-rank compression techniques in most instances, the paper provides insufficient explanation for instances in which A^3’s performance falls below that of its counterparts.  For instance, in Table 2, under certain configurations, SVD-LLM outperforms A^3 on nearly every benchmark.  Some explanation is provided in Appendix D, suggesting that SVD-LLM’s superior performance may be a function of overfitting, but no explanation is provided for why overfitting should result in superior performance across multiple distinct benchmarks.

---

> ### Author Rebuttal · Authors · 2026-03-30
>
> Thank you for the careful review and enlightening questions! Here we answer them one by one.
>
> > W.1
>
> While $A^3$ is inspired by output-error minimization, it differs fundamentally in problem formulation and solution and practical capabilities.
> ### **Different Optimization Objective and Solution: Joint Optimization vs. Single-layer Optimization**
> **1. Joint vs. layer-wise optimization**
> * **SVD-LLM:** Optimizes each layer independently
>   $$
>   \min_{A,B} |X(W - AB)|
>   $$
> * **$A^3$:** Jointly optimizes attention + MLP components
>   $$
>   |A-\tilde A|,; |O-\tilde O|,; |MLP-\widetilde{MLP}|
>   $$
>
> **This joint optimization:**
> * **More accurate modeling of end-to-end behavior**. Unlike SVD-LLM that minimises the output error of individual linear layers, our formulation approximates the joint behavior of multiple layers, providing a much closer proxy of model’s end-to-end performance.
> * **Zero-overhead structure:** $A^3$ reduces native model dimensions, so compression **directly translates** into proportional speedup, KV-cache reduction, and memory savings. In contrast, SVD-LLM’s rank compression does not map directly to FLOPs/memory gains and introduces extra GEMM kernel launches due to splitting linear layers, reducing practical throughput (see Fig. 5, Table 12).
> * **Improved attention modeling beyond KV-cache compression methods.** Although $A^3$ is a whole-model compression method, its objective captures both **attention weights and attention outputs**. This leads to results that outperform even recent KV-cache specific methods, Clover and Palu. In Table.3 of the paper we highlight KV cache compression analysis on MPT-7b and 30b.
>
> **2. Practical innovations**
> Modern transformers utilize NoPE, RoPE, MHA, GQA and three layer MLP. $A^3$ supports all of them, while Palu and Clover don't support RoPE.
> > W.2
>
> Yes we didn’t provide theoretical analysis for this as analyzing intermediate optimization objectives in black-box models is very challenging. Below is our assumption and **we test it via extensive experiments.**
> We view objective design as progressive: layer-wise → functional components → end-to-end output. $A^3$ better approximates global behavior by minimizing functional component errors.
> Appendix D further provides diagnostics linking **local objective** (e.g., QK, OV) to **end-to-end perplexity**. We will clarify this limitation in the paper.
> > W.3
>
> Low-rank approximation and quantization are generally considered distinct but complementary methods, each with different trade-offs. Appendix D discusses their integration with $A^3$. Empirically, Table 10 shows that combining them improves compression: from 4× (4-bit quantization) to 5× with (4-bit+$A^3$@0.8), and up to 6.67× with (4-bit+$A^3$@0.6).
> 1. Advantages of Quantization
> Naive low-rank methods often underperform quantization for bandwidth reduction. For instance, 4-bit Quarot achieves 6.1 perplexity (~75% compression), while SVD-LLM yields 66.62 at 60% and 7.94 even at 20% compression.
> 2. Advantages of Low-Rank Decomposition
> However, quantization and low-rank decomposition are not directly comparable, as low-rank methods offer distinct advantages:
> * **Hardware independence:** Quantization often needs specialized hardware (e.g., A100/A6000: BF16/FP16/INT8/INT4) to achieve real speedups; otherwise, dequantization adds computational overhead. Low-rank methods deliver both memory savings and throughput gains on any hardware.
> * **Finer control & extreme compression:** Low-rank + quantization enables continuous compression between discrete bit levels, improving the Pareto trade-off. As shown in Fig.5, Pure sub-3-bit quantization is unstable, while $A^3$ + quantization provides a significantly better Pareto trade-off.
> * **Complementary to quantization:** Many works (e.g., SVDQuant, QERA, Slim, LQER, OATS) combine both Low-rank and quantization for superior overall compression.
> > W.4
>
> Cases where $A^3$ underperforms SVD-LLM (e.g., Table 2) are mainly due to calibration sensitivity and CUR decomposition limitations, both discussed in the Limitations section and further analyzed in Appendix D.
> Table 3 reports results on MPT-7B and MPT-30B across compression rates from 20% to 80%, excluding CUR decomposition. In this setting, $A^3$ achieves up to an order-of-magnitude improvement over prior methods Clover, Palu. We reveal a key structural insight: for MHA layers without RoPE, the QK and OV components are highly low-rank, and models remain functional even after removing up to 50% of the head dimension. We believe this finding has broader implications(e.g model design) beyond compression. Future directions in low-rank could investigate new formulation to improve on this.
> > Q.1
>
> Yes, we will revise the manuscript to include the subscripts to reflect your changes.
>
> > Q.2
>
> You are correct. U determines which rows of W_d and which columns of X_d are selected. We will revise the manuscript to correct this and clarify the description.
> > Q.3
>
> Please see W.2 reply.

---

> > ### Author Rebuttal · Reviewer_hk78 · 2026-04-03
> >
> > Thank the authors for the rebuttal. After reviewing, I'd like to keep my original score.

---

> > > ### Author Response · Authors · 2026-04-05
> > >
> > > Thank you for confirming that our rebuttal addressed your concerns. We will incorporate these clarifications to strengthen the paper. Is there anything we could further address that would increase your score towards the paper?

---

### Decision · Program_Chairs · 2026-04-30

**Decision:**

Accept (regular)

**Comment:**

This paper proposes A^3, a post‑training low‑rank compression framework that approximates native Transformer components (QK, OV, and MLP) rather than individual linear layers, enabling direct reduction of hidden dimensions without introducing inference‑time overhead. Reviewers agree the method is technically sound, well presented, and offers a meaningful architectural improvement over traditional layer‑wise low‑rank approaches, particularly in avoiding extra GEMMs and enabling KV‑cache compression. The analytical formulation and extensive evaluations across model sizes are strengths.

At the same time, reviewers consistently note limitations in practical impact: performance degrades noticeably beyond modest compression rates (<20%), applicability is constrained by architectural specifics, and gains as a standalone method may not justify replacing smaller base models. The method appears most valuable as a complementary building block, especially when combined with quantization or fine‑tuning, rather than as a complete compression solution.

Overall, the work presents an interesting and solid contribution with clear technical merit but moderate scope. I recommend Weak Accept.